# Challenges in Media Attention toward COVID-19-Preventive Behaviors: Dual Roles of Threat and Perceived Capability for Information Systems and Health Care

**DOI:** 10.3390/bs14050377

**Published:** 2024-04-30

**Authors:** Suyu Chou, Rebecca Katherine Britt

**Affiliations:** College of Communication & Information Sciences, University of Alabama, Tuscaloosa, AL 35487, USA; rkbritt@ua.edu

**Keywords:** health care, information systems, perceived risk, social networks, legacy media, new media

## Abstract

Based on the extended parallel process model, this study investigated the relationship between young adults’ media exposure to COVID-19 and their adoption of protective behaviors. This study surveyed 141 college students and found that increased risk perceptions led to greater intentions to engage in COVID-19-preventive behaviors and that these intentions were mediated by normative beliefs. There was no significant difference in risk perceptions between traditional media and social media. The results showed that college students took precautions against COVID-19 because they perceived themselves to be both vulnerable and capable.

## 1. Introduction

COVID-19, characterized as a respiratory illness, is triggered by a novel coronavirus previously unidentified. The U.S. Food and Drug Administration (FDA) in 2022 highlighted its unprecedented nature, marking a significant challenge for global health systems. Originating in China in December 2019, this pandemic has since unfurled globally, affecting millions. As of 18 February 2024, the World Trade Organization (WTO) reports a staggering 774.6 million confirmed cases and an approximate death toll of 7 million individuals worldwide [1]. This pandemic underscores not only the transmissibility of the virus but also the immense impact on public health, economies, and daily life across the globe.

Historically, humanity has been confronted with numerous outbreaks of viral illnesses that have threatened lives and incited widespread panic. Notable instances include the Ebola outbreaks of 1976 and 2014, the Severe Acute Respiratory Syndrome (SARS) outbreak in 2003, the H1N1 influenza pandemic in both 1918 and 2009, and the Middle East Respiratory Syndrome (MERS) emergence in 2012. These outbreaks have starkly reminded us of the devastating impact that viruses can have.

In response to such health crises, research conducted by Lin et al. emphasizes the crucial role of governmental communication strategies in mitigating public panic and promoting safety measures. Effective communication and public health initiatives have been recognized as essential tools in managing the spread of diseases, facilitating public understanding of risks, and implementing necessary precautions to protect public health [2].

During times of health crises, the general people tend to seek reliable information and support to navigate the uncertainties posed by novel diseases. The ambiguity surrounding new pathogens often leads to increased anxiety and distress among individuals. In this context, well-coordinated public health communication strategies are vital. They not only inform and educate the public but also play a pivotal role in alleviating fears by providing clear, accurate, and timely information. This, in turn, helps in fostering a sense of community solidarity and collective action in facing health challenges.

The purpose of this study is to better understand how college students’ risk perception of COVID-19 from the media and their media attention directly influence their protective behaviors during the COVID-19 pandemic. Recent studies indicate that how people use social media is related to risk perceptions of health issues [2,3,4,5]. The extent to which legacy media shapes people’s perception of risk is currently being studied and examined. Chan et al. demonstrate that both traditional media and new media are important information sources when the public encounters emerging health risks [6]. Our research sought to determine if exposure to various forms of media (traditional and social) results in varying levels of risk perception toward COVID-19. 

Guided by the extended parallel process model (EPPM), this research focuses on college students’ media attention on COVID-19 and their intention to take protective health behaviors. Given that social interactions are often of great importance to college students, we considered the impact of social norms as a mediator variable to investigate whether their behavior is influenced by normative beliefs. The results of this current study illuminate how college students evaluate COVID-19-related risk from media and how they protect themselves from contracting COVID-19. Therefore, the results provide an understanding of how the effectiveness of risk message design in health campaigns and interventions can serve as information to public health communicators.

## 2. Literature Review

### 2.1. EPPM

Understanding how individuals perceive risks and make behavioral changes is crucial in various contexts, particularly in health communication [7]. The extended parallel process model (EPPM) is a pivotal framework that delves into this relationship by examining how risk perceptions and fear appeals influence decision-making processes [8,9]. In addition, the EPPM takes into account both the cognitive and emotional components of message processing, offering a comprehensive approach to understanding how individuals react to threats and advisories [7]. The conceptual model of the EPPM is illustrated in Figure 1 [10]. The EPPM has undergone extensive development and testing throughout the years, until today, since it was first developed by Kim Witte in 1992. It is especially widely used in health communication research to design and evaluate public health campaigns [11,12]. Multiple studies have identified a significant, linear, and positive relationship between fear arousal and persuasive outcomes, such as attitude and behavior changes [13,14,15,16].

The EPPM explains how emotional arousal (fear) and cognitive capacity combine to influence behavioral decisions [7]. The theory states that when people perceive a threat, two major factors in message processing, perceived efficacy and perceived threat, determine how people elaborate on the risk. According to Witte, a perceived threat is made up of two components: perceived severity and perceived susceptibility [7]. This is based on Leventhal’s parallel process model, which extends the danger/fear control framework and fear appeal [17] with elements of Roger’s protection motivation theory [18,19]. For instance, in the current study, the perceived severity is defined as “It is likely that I would be affected by COVID-19” and the perceived susceptibility as “I believe being infected by COVID-19 is significant, that might lead to death”.

The efficacy is discussed in two dimensions: response efficacy and self-efficacy. According to Witte, response efficacy is the conviction that the suggested solution is effective in thwarting the threat [7]. Self-efficacy is characterized as beliefs about one’s capacity to carry out the advised action to counter the threat (for example, “I believe adopting protective behaviors is useful to prevent COVID-19 contraction”). A person adopts the message if they recognize the threat and have high efficacy following the EPPM. In contrast, even if a risk is highly powerful, people do not feel driven to take preventive measures if they do not see it as a threat to them. According to Barnett, those who perceive a high threat and high efficacy are more likely than those who perceive a low threat and low efficacy to adopt suggested pandemic flu prevention measures [20]. Shirahmadi et al. researched oral health care professionals using the EPPM and found that preventive actions such as routinely washing hands with soap and using hand sanitizer are substantially associated with high perceived danger and efficacy [9]. According to the EPPM, perceived threat and efficacy are sufficient to spur college students’ preventive measures in the face of the COVID-19 pandemic.

### 2.2. Risk Perception

The authorization of COVID-19 vaccines in April 2021 marked a significant milestone in the fight against the pandemic, leading to a notable shift in public opinion toward mitigation efforts. This change has been particularly evident among college students, a demographic known for their eagerness to embrace new life experiences, including adapting to the evolving landscape of the pandemic. Despite this enthusiasm, the importance of not sidelining protective measures against COVID-19 cannot be overstated. In this context, how young adults process and respond to COVID-19-related messaging plays a pivotal role, which not only influences their readiness to adopt protective behaviors but also affects the extent of their compliance with the guidelines set forth by educational institutions and health authorities.

This study aims to delve into the intricacies of how young adults engage with media narratives surrounding COVID-19, their risk perceptions, and the subsequent impact on their protective behaviors. Research by Rubin et al. underscores the correlation between the level of public awareness of a disease and the adoption of recommended safety practices [21]. The way critical pandemic information is communicated through various media outlets is instrumental in guiding individuals toward effective self-protective actions and in reducing the transmission rates within communities. Therefore, understanding young adults’ perceptions of the threat posed by COVID-19 becomes essential in devising strategies for effective disease prevention and management.

### 2.3. Normative Beliefs

The COVID-19 pandemic has transformed the dynamics across all aspects of life, with campus life for both students and educational institutions being no exception. Amid this global health crisis, individuals may feel a heightened sense of obligation to conform to COVID-19 protective measures, such as participating in public discussions or assuming social responsibilities, driven by the imminent threat posed by the virus. In this context, normative beliefs play a crucial role as a mediating variable in understanding the behavioral responses to the pandemic. Given that young adults place a high value on social connections, the influence of these beliefs on adherence to protective behaviors is likely to be pronounced. Therefore, peer influence, reinforced by normative beliefs, is anticipated to significantly motivate students to engage in recommended protective practices.

Normative beliefs are essentially individuals’ perceptions of the social pressures or expectations they believe are held by significant others or society at large regarding how they should behave. According to Ajzen’s Theory of Planned Behavior (Ajzen, 1991) [22], normative beliefs, together with attitudes toward the behavior and perceived behavioral control, contribute to the formation of behavioral intentions, which in turn predict actual behavior. In the context of the COVID-19 pandemic, these beliefs may encompass expectations about wearing masks, social distancing, and hand hygiene, reflecting the perceived approval or disapproval of these actions by peers, family members, or the community.

This study employs the EPPM and the concept of normative beliefs to evaluate protective behaviors against COVID-19. The EPPM suggests that the way individuals process and respond to threatening information is influenced by their perceived severity and susceptibility to the threat, as well as their perceived efficacy in taking actions to mitigate the threat [7]. By integrating normative beliefs into this model, we aim to gain a deeper understanding of how societal expectations influence individual decisions to adopt or reject protective measures. Normative beliefs, as defined in this study, refer to individuals’ interpretations of the normative expectations they perceive to be endorsed by society, influencing their compliance with public health guidelines during the pandemic.

In summary, normative beliefs, representing societal and peer expectations, could be crucial in influencing how individuals react to the COVID-19 pandemic, especially young adults within higher education environments. By examining the interplay between normative beliefs and protective behaviors through the lens of the EPPM, this study seeks to uncover the mechanisms by which social norms influence adherence to health-protective measures, offering insights into effective strategies for promoting public health compliance in the face of ongoing health threats.

### 2.4. Legacy and Social Media

Social media is a common way to acquire news in our modern media landscape. According to the Pew Research Center, more than two-thirds of Americans use social media to communicate with others and learn about current events [23]. Organizationally, the Centers for Disease Control and Prevention (CDCs) frequently make use of social media to enhance public awareness of the pandemic and inform people (e.g., about the Zika and Ebola outbreaks) [24,25]. The public is inclined to communicate their health concerns to seek medical assistance due to the often-decentralized nature of platforms [25,26]. Allen et al. used geographic information science (GIS)-stored algorithms through Twitter to track influenza epidemics, suggesting that tracking disease control over time could aid in predicting and tracking disease outbreaks [27].

Public responses to breaking news on social media during an outbreak of disease may increase anxiety in communities [28,29]. Chan et al. found that whereas traditional media coverage was favorably connected to people’s preventive behaviors, social media coverage was positively related to people’s perceptions of risk [6]. A primary objective of the current study is to determine how behavioral intentions may be influenced by the massive volumes of COVID-19 information published on social media. As a result, social media and legacy media represent the two categories in which media attention in this study is divided.

## 3. Methods

The current study uses the EPPM with normative beliefs as a mediating variable to determine key factors that influence people’s risk perceptions with COVID-19 and how those characteristics relate to intention and actual preventive behavior among young adults (Figure 2). The COVID-19 pandemic is a persistent global health crisis, making it an ideal opportunity to investigate the relationship between behavioral intentions and practical behaviors. In addition, we investigate several demographic and health status characteristics to see if they influence young adults’ compliance in undertaking COVID-19 preventive behaviors. The following research questions and hypotheses are addressed.

**H1:** 
*Increased risk perceptions will result in greater intentions to take preventive behaviors against COVID-19.*


**H2:** 
*Greater intentions to take preventive behaviors against COVID-19 will result in greater protective behaviors and will be moderated by normative beliefs.*


***RQ1:*** 
*What are the levels of risk perceptions of COVID-19 from legacy media and social media? (Do people who obtain their information about COVID-19 from social media, compared with legacy media, have increased risk perceptions?)*


***RQ2:*** 
*How will risk perception types (high threat/high efficacy, high threat/low efficacy, low threat/high efficacy, and low threat/low efficacy) result in different levels of discrepant intentions to take preventive behaviors against COVID-19?*


***RQ3:*** 
*What are the risk perceptions and protective behaviors of individuals who have COVID-19, compared to people who do not have COVID-19?*


***RQ4:*** 
*What demographic factors influence the risk perceptions and protective behaviors about COVID-19?*


### Participants and Procedure

With IRB consent obtained, participants were recruited from a southern university in the United States. The students completed the study through an online recruitment system. Upon completion of the study, they received a small portion of credit toward a class for their time. A total of 141 participants completed the survey. The survey was conducted from 16 February to 27 October 2021, a prime period for assessing vaccine acceptance. This timeframe coincided with the global rollout of vaccines by Pfizer-BioNTech, Moderna, and AstraZeneca, following their emergency use authorization in December 2020 [30]. Notably, the FDA approved the first COVID-19 vaccine on 23 August 2021 [31]. The survey’s timing was thus crucial for gauging public response soon after the vaccines became available.

The items utilized a 5-point Likert scale to assess variables. Details on the structure of the survey questions are provided in Table 1. To address H1, a linear regression was conducted. H2 was tested using linear regression and PROCESS for mediation analysis. RQ1 was examined using an independent-sample *t*-test. RQ2 was addressed using a one-way Analysis of Variance (ANOVA). RQ3 was addressed using group means compared using an independent-samples *t*-test. RQ4 was answered using an independent-samples *t*-test and a one-way ANOVA.

## 4. Results

Of the 141 participants, 19% were male, and 81% were female. A total of 88% were White, 6% were Black, and 5% were Hispanic. While 70% of participants received COVID-19 information from social media, 30% received information from legacy media. As for the question “What is your current status regarding COVID-19?”, 29% of participants claimed that they had tested positive for COVID-19 and recovered, while 71% were negative.

A linear regression analysis was conducted to test H1. The regression analysis suggested a significant association between risk perceptions and protective behaviors. F (1, 139) = 106.404, *p* < 0.001, adjusted *R*^2^ = 0.430, using the following equation: Y = 0.352X + 1.679. H1 was supported.

To test H2, a linear regression analysis was conducted. The regression analysis suggested a significant association between protective intentions and protective behaviors. F (1, 139) = 116.675, *p* < 0.001, adjusted *R*^2^ = 0.452, Y = 0.568X + 4.682. Based on PROCESS, protective intentions can significantly predict protective behaviors, t = 10.802 and *p* = 0.000. The total effect is 0.568. The indirect effect is 0.0135 with a 95% CI of [0.0415, 0.2502]. Additionally, 23.8% of the total effect is mediated by the mediator (normative beliefs). Moreover, 45.63% of the variation in protective behaviors can be explained by protective intentions, whereas 26.91% can be explained by the mediation effect. H2 was supported.

To address RQ1, the group means were compared using an independent-samples *t*-test. The average scores for COVID-19 risk perceptions were slightly higher for those who obtained news from social media (M = 22.26 and SD = 5.54) compared to obtaining news from legacy media (M = 21.81 and SD = 5.55). However, the results show that there is no significant difference in risk perceptions between different media types, t (139) = 0.435 and *p* > 0.05.

To address RQ2, four scenario-specific categories for the EPPM were created, based on the level of the perceived threat and perceived efficacy. These categories included low threat and low efficacy (LT/LE), low threat and high efficacy (LT/HE), high threat and low efficacy (HT/LE), and high threat and high efficacy (HT/HE). Using Likert scale responses, the “threat” variable was determined as the product of the participant’s response to the perceived likelihood of the COVID-19 threat and the perceived severity of it. In contrast, the “efficacy” variable was calculated as the product of the participant’s response to their perceived ability against COVID-19. Low and high categories of the perceived threat and efficacy were determined by the median value of each product, respectively.

The group means were compared using a one-way ANOVA. The results reach significance in different risk perception types of COVID-19 (Table 2).

To answer RQ3, the group means were compared using the independent-sample *t*-test. The average score of risk perception was slightly higher among college students who were COVID-19 negative than the score of COVID-19-positive students. Their answers indicated that COVID-19-negative students were more worried about COVID-19 (M = 22.3 and SD = 5.3) than college students who had tested positive (M = 21.6 and SD = 6.2), although there was no significant difference between the positive and the negative. Likewise, from the average score, COVID-19-negative students believed taking protective behaviors helped prevent COVID-19 contraction (M = 8.4 and SD = 1.9) more than college students who had tested positive for COVID-19 (M = 7.9 and SD = 2.6), yet there was no significance difference between the positive and the negative.

To address RQ4, an independent-samples *t*-test and a one-way ANOVA were conducted. There was no significant difference in risk perceptions and protective behaviors between gender and living status (on-campus or off-campus). There was a difference in risk perceptions for students who currently had a part-time job (M = 24.03 and SD = 4.10) and those who did not have a part-time job (M = 21.37 and SD = 5.85), t (139) = 2.63 and *p* = 0.010 < 0.05. In terms of risk perceptions, the results showed a significance at different levels of physical health (F (1,139) = 6.63 and *p* = 0.011 < 0.05) with an adjusted *R*^2^ of 0.011 and in different levels of mental health (F (1,139) = 7.06 and *p* = 0.009 < 0.01) with an adjusted *R*^2^ of 0.041. On the other hand, regarding protective behaviors, there was no significant difference in either physical health or mental health.

## 5. Discussion and Conclusions

The findings of this study demonstrate how perceived efficacy and perceived threat levels influence young adults’ willingness to engage in advised protective behaviors. Elevated risk perceptions lead to higher intentions to adopt COVID-19-preventive measures, which means when young adults perceive a higher level of risk, their motivation to adopt measures preventing COVID-19 significantly increases. Furthermore, greater protective behaviors are induced by stronger intentions to take preventive measures against COVID-19, and normative beliefs regulate this relationship. This indicates the degree to which college students care about other people’s opinions as well as their sense of risk being associated with their protective behaviors. This study suggests that college students’ protective actions are deeply intertwined with their concern for others’ perceptions and their assessment of the dangers posed by COVID-19. This finding opens new avenues for future research, particularly in examining other demographic groups that might be more susceptible to peer influences. By leveraging the extended parallel process model (EPPM) and incorporating normative beliefs, future studies can gain insights into how different populations navigate the complexities of health-related decision-making in the face of pandemics. This approach could be especially beneficial in designing more effective health communication strategies that resonate with diverse audiences, ultimately encouraging wider adoption of protective measures.

Interestingly, our analysis revealed no noticeable distinction between social media and legacy media concerning the types of media and their impact on risk perceptions of COVID-19. This outcome suggests that the content of messages carries more weight than the platforms on which they are disseminated, even though social media serves as the primary news source for 70% of the respondents. Consequently, it appears that beyond the sheer diversity of media platforms available, other elements such as how messages are framed or the frequency with which individuals are exposed to media content could significantly affect their perceptions of personal risk. Therefore, additional research is necessary to acquire a more comprehensive understanding of these dynamics. This further exploration could unravel how different media types and their specific characteristics influence public perceptions and behaviors in the context of a global pandemic.

The findings also reveal a significant variation in protective intention and protective behaviors across the four degrees of perceived threat and perceived efficacy. According to this study’s findings, participants who perceived a high threat and high efficacy had the highest protective intention scores on COVID-19, and their scores differ significantly from those of all other participant categories. Participants who perceived a high threat and efficacy also scored highest on protective behaviors related to COVID-19, but only one group with low threat and efficacy showed a meaningful difference in terms of actual protective actions. The idea that individuals with low threat and efficacy had the lowest intention to engage in preventive activities can be used to explain the overall findings. On the other hand, those with high threat and high efficacy have the highest propensity to act. Other circumstances, including high threat and low efficacy and low threat and high efficacy, do not significantly differ in terms of the defensive behaviors taken. As a result, in determining protective behavior, both threat and efficacy play a significant role. University students adopt preventive measures against COVID-19 due to their perception of both the severity of the threat and the effectiveness of the precautions. These findings underscore the critical interplay between perceived threat and efficacy in shaping individuals’ intentions and actual behaviors toward COVID-19 preventive measures. Specifically, the results highlight that university students are more inclined to adopt protective actions when they recognize the severity of the threat and believe in the effectiveness of the precautionary measures available. Consequently, it becomes paramount for health communication strategies to focus on enhancing both the perceived threat of COVID-19 and the perceived efficacy of protective behaviors to effectively motivate a higher degree of protective action among the population.

Those who tested negative for COVID-19 held higher scores on risk perceptions and protective behaviors in preventing COVID-19 transmission than those who tested positive for the virus. However, the results did not show a significant difference in these behaviors between the COVID-19-positive and -negative students. Those who were employed part-time had a greater COVID-19 risk perception than those who were not. The perception of COVID-19’s danger is significantly influenced by both physical and mental health, so the healthier a participant is, the more conscious of COVID-19 they are. Different levels of physical and mental health do not, however, significantly differ in terms of protective actions. While higher risk perceptions and protective behaviors were observed among those who tested negative for COVID-19 and those employed part-time, these variations did not consistently translate into significant differences in protective actions across different health levels or COVID-19 test outcomes. This indicates that while awareness and perceptions of COVID-19 risk may be influenced by an individual’s health and employment status, these factors alone do not determine the extent of protective measures undertaken, highlighting the complexity of factors driving pandemic-related behaviors.

A major limitation of this study is its relatively small sample size of 141 participants, which casts doubt on the generalizability of the findings. Furthermore, the participants’ demographic distribution, with 19% men and 81% women, lacked balance. Future data collection should aim for a more diverse range of demographic characteristics to enhance representativeness. Moreover, the use of a self-reported questionnaire to gather data introduces the potential for social desirability bias, wherein participants may report behaviors that reflect societal expectations rather than their actual practices. This aspect further complicates the interpretation of the study’s outcomes, underscoring the need for more comprehensive methodologies that can minimize bias and provide a more accurate reflection of individual behaviors. Expanding the participant base and ensuring a more balanced gender representation in future studies could be one approach to enhancing our understanding of the dynamics at play and offering more robust conclusions.

This study provides empirical support regarding the relationship between social and traditional media and risk perceptions, normative beliefs, protective intentions, and protective behaviors about COVID-19. To summarize the key findings of this paper, it is unsurprising that participants who perceived both high threat and high efficacy displayed the highest protective intentions and behaviors toward COVID-19. However, the significant influence of normative beliefs on risk perception and behavior suggests that both societal opinions and college students’ own risk assessments shape their protective actions. This highlights the potential for “normative beliefs” to expand the EPPM. Future research should delve into the origins of normative perspectives, such as individualism or collectivism, to further refine the EPPM framework. Understanding that both societal influences and personal risk assessments impact behavior can guide the development of more tailored risk communication strategies. For example, public health messages can be designed to align with prevalent normative beliefs within specific communities, enhancing their effectiveness. Additionally, acknowledging the influence of individualism or collectivism could lead to culturally adapted communication strategies that resonate more deeply with different demographic groups, thereby increasing compliance with health advisories. Policymakers can leverage these insights to craft policies that not only disseminate information but also engage community norms and values, fostering a more cooperative public response. This approach could be particularly effective in managing and mitigating the impact of future health crises by ensuring that communication strategies and policies are both culturally sensitive and scientifically sound.

## Figures and Tables

**Figure 1 behavsci-14-00377-f001:**
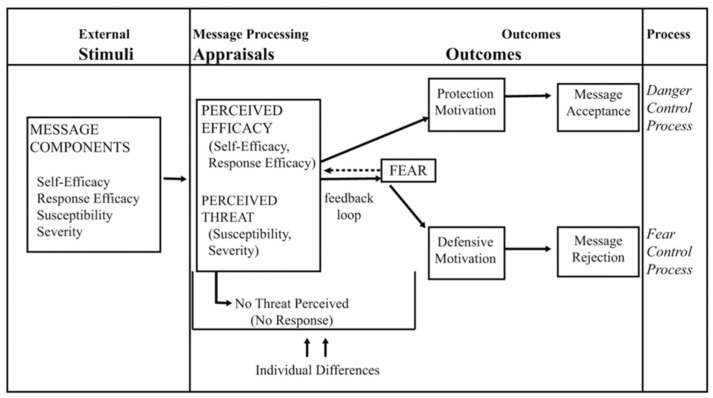
The EPPM model. Source: Witte (1996) [10].

**Figure 2 behavsci-14-00377-f002:**
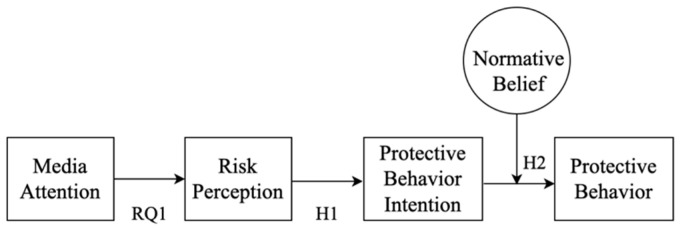
Theoretical research model.

**Table 1 behavsci-14-00377-t001:** Items for measuring variables: results, means, and standard deviations.

Variable	Item	Result, Mean, SD		
Media Attention	Where do you get the news of information related to the COVID-19 pandemic?	Social M	70%	
		Legacy M	30%	
COVID-19 Status	What is your current status regarding COVID-19?	Positive	29%	
		Negative	71%	
			M	SD
Threat	I am worried that I would be affected by COVID-19 again	Positive	2.7	1.2
	I am worried that I would be affected by COVID-19	Negative	3.0	1.4
	It is likely that I would be affected by COVID-19 again	Positive	3.4	1.1
	It is likely that I would be affected by COVID-19	Negative	3.3	1.2
	The problem of the COVID-19 pandemic is serious to me.		3.8	1.2
	I feel that COVID-19 is dangerous.		3.9	1.2
Efficacy	I still believe taking protective behaviors is helpful to prevent COVID-19 contraction.	Positive	3.8	1.4
	I believe taking protective behaviors is helpful to prevent COVID-19 contraction.	Negative	4.1	1.1
	I still think I can take protective behaviors to prevent COVID-19 contraction.	Positive	4.0	1.2
	I think I can take protective behaviors to prevent COVID-19 contraction.	Negative	4.3	1.0
Social Norm	How responsible do you feel about your own role in preventing further outbreak of the COVID-19 pandemic?		3.1	1.0
	I care about others judgement if I do not actively take actions against the COVID-19 pandemic such as wearing the mask?		3.2	1.3
Intention	I have intended to wear a mask to reduce the risk of COVID-19 infection.		4.3	1.1
	I have intended not to go to public spaces, such as restaurants or department stores.		2.1	1.2
	I have intended to keep social distance to reduce the risk of COVID-19 infection.		3.1	1.4
Behavior	I have worn a mask to reduce the risk of COVID-19 infection.		3.9	1.0
	I have avoided social gatherings with friends.		2.1	1.1
	I have tried to wash my hands or used hand sanitizer more often to reduce the risk of COVID-19 infection.		4.1	1.1

**Table 2 behavsci-14-00377-t002:** ANOVA results of risk perception types on protective intention and behaviors.

Protective intention	(F (3,137) = 27.874, *p* = <0.001)
Protective behaviors	(F (3,137) = 27.048, *p* = <0.001)
Post hoc high threat and high efficacy (HT/HE)	(M = 11.54, SD = 2.62) (HT/LE) (M = 8.84, SD = 1.95) (LT/HE) (M = 9.11, SD = 2.05) (LT/LE) (M = 7.20, SD = 2.32) (Tukey’s HSD, *p* ≤ 0.001)
Protective behaviors	(HT/HE) (M = 11.59, SD = 1.99) (LT/LE) (M = 7.93, SD = 2.11) (Tukey’s HSD, *p* ≤ 0.001)

## Data Availability

The data are contained within this article. The data used in this research are available via the Open Science Framework (OSF), https://osf.io/8du3f/ (accessed on 29 November 2023).

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
