# Peer review of "Challenges in Media Attention toward COVID-19-Preventive Behaviors: Dual Roles of Threat and Perceived Capability for Information Systems and Health Care"

_behavsci, 2024, doi:10.3390/bs14050377_

Round 1

Reviewer 1 Report

Comments and Suggestions for Authors

A visual of the EPPM model would have been useful. On page 2, section 2.1 "throughout the years" could be strengthened by actually citing specific names that would be recognizable to the audience. This would be a better connection to prior research than saying "throughout the years"; On that same page, line 86-87 the wording of the last quote's language needs to be reworded so the sentence is clear.  When the writers use embelled phrases like "profoundly transformed" the question arises as to what the authors mean in the context of the research.  What makes something "profoundly transformed" especially in the context of college campuses. College campuses where? Authors cannot speak to all college campuses because of the diverse responses utilized by colleges across the globe. This is a case of avoiding broad generalizations and flowery, puffy language.  I think the top of page 4.10 was well said and clearly stated in regard to the authors' aims. The area of concern I have for the article is the lack of specific nuanced explanations of methods. I understand there was a survey, but the authors provide no examples of the types of questions asked in the survey. They do not explain how long the survey is nor, when we get to results do we get any indication of what the responses to the survey. The statistcal data might be worthwhile, but without a sound context, it falls short. The methods section needs a bit more development which in turn would lead to more development in results and discussion. In the results and discussion the authors note the male to female ratio; but, in contemporary practice I think scholars would also be interested in the data from the demographic subset related to gender/sexual identity. College campuses are almost like microcosms of the larger society. I don't think the point has to be belabored, but I do think the binary male/female is problematic. If the authors limited their question by asking a person to identify their biological/birth gender, that might be different. Limiting the question to male/female though and then wondering why there were so many more women than men, bares considering how that question was phrased and answered.

Comments on the Quality of English Language

In section 2.2 and throughout avoid starting sentences with "it". In English, "it" can be a very vague pronoun and can lead to confusion for the reader  because the reader has to decide which 'it' the authors are referencing. In the instance in section 2.2, the 'it' is unclear.

Reviewer 2 Report

Comments and Suggestions for Authors

#1: This article provides further insight into the theories that support the conceptual framework of the study and contribute to the field. 

It would be beneficial to present a sample that reflects a more diverse range of gender, achieving a more equal distribution, or in terms of ethnicity or other relevant factors. This could be useful in future studies. 

#2: I call for further discussion of the practical implications. It would be useful to include an additional section elaborating on how research findings can inform specific risk communication strategies and public health policies. 

It would also be useful to expand the bibliography.
